# The Influence of Meteorological Conditions on the Yellow Fever Epidemic in Cádiz (Southern Spain) in 1800: A Historical Scientific Controversy

**Fernando S. Rodrigo** 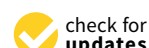

Department of Chemistry and Physics, University of Almería, 04120 Almería, Spain; frodrigo@ual.es

**Abstract:** A yellow fever epidemic occurred in Cádiz and other areas of southern Spain during the last months of 1800. An anonymous author attributed this disease to the contrast between the cold and rainy winter and spring, and the subsequent very hot summer. However, the physician J.M. Aréjula published a report in 1806 where he refuted this conclusion after a detailed analysis of the meteorological conditions in the area. This controversy is a good example of the discussion about the relationships between meteorological conditions and public health. In this work, this "scientific" controversy is studied. Although the arguments of both authors were inspired by the neo-Hippocratic medical paradigm, the anonymous author put forth a simple cause–effect hypothesis, while Aréjula recognized the complexity of the problem, introducing the concept of "concause" to explain the confluence of environmental and contagious effects.

**Keywords:** climate and health; yellow fever; Spain; historical climatology

## 1. Introduction

Global climate change has suggested that a re-evaluation of the role of the climate in human health is needed [1]. The influences of weather and climate on human health are significant and varied. They range from the clear threats of meteorological extremes to connections that may seem less obvious, such as the survival distribution and behavior of mosquitoes that carry diseases [2]. These vectors are sensitive to changes in climate conditions, especially temperature and humidity [3].

Yellow fever (YF) is a disease transmitted to humans by the bites of the *Aedes aegypti* mosquito, where an infected individual is bitten by a single species of mosquito, and, after an incubation period, the mosquito bites another individual, passing the pathogen from one person to another. Climate variability largely determines the distribution and population dynamics of *Aedes aegypti* on a global scale [4]. *Aedes eagypti* is typically found in tropical and subtropical regions worldwide within urban areas, where it can exploit water-filled containers for its immature (larval and pupal) stages [5]. The rate of growth of a mosquito population is dependent on the initial population size before the rain season. Rainfall increases the availability of mosquito breeding habitats and thus the size of the mosquito population. Temperature controls the rate of larval development. Higher temperatures shorten the development time of the larvae in the mosquitoes. The optimum temperature for the functional activity, breeding, and feeding of *Aedes aegypti* is from 27 to 31 °C [6] Therefore, climatic factors may contribute to severe YF outbreaks by promoting the reproduction, survival, and propagation of the *Aedes aegypti* mosquito. The intensity of the transmission is proportional to the size of the mosquito population [7]. Local climatic parameters play a central role in determining the distribution and abundance of vector organisms through the effects on the host animals. Therefore, it is anticipated that global climate change will have significant effects on the geographical range and seasonal activity of many vector species [1–3]. In particular, the mosquito *Aedes albopictus* is the vector that causes the transmission

of chikunguya virus and dengue virus, and it is expected that an increase in climatic suitability may occur in many European areas in a warmer world, although some uncertainties related to precipitation persist [8]. Due to Spain being located near Africa, being a stopping-off point for migrating birds and individuals, and due to its climate conditions, nearing those of areas where there are vector-borne diseases, this is a country where this type of disease could take on greater importance due to climate change; the possible risk would result from the geographical spread or adaptation of vectors [9].

Human health has always been influenced by climate and weather [2]. In past agrarian societies, one of the consequences of climate anomalies was the emergence of diseases and epidemics, along with their social and economic aftermaths [10]. A shortage of food resulting from extreme events, such as droughts and floods, contributes to malnutrition and weakened immune systems, resulting in ill health, which makes individuals easily succumb to diseases [6]. The complex interaction between climate variability and human history has attracted renewed interest in academic and popular literature [11]. Documenting past climate variability and its potential impact has become an interesting focus of paleoclimate and historical research and may help to prepare modern society to cope with the probable anthropogenic climate changes of the present. Some studies have analyzed the role of climate variability in historical diseases and epidemics [10,12,13], but these types of studies (see, for instance, [14]) are scarce in the case of Spain.

A YF epidemic occurred in Cádiz and other areas of southern Spain during the last months of 1800. The present study considers climate conditions from 1799 to 1800 in Cádiz (southern Spain), which may have contributed to enhancing the mosquito population during the 1800 summer season. It was not until the end of the nineteenth century that the apparently high number of mosquitoes noted during many YF outbreaks was connected to the disease [15]. The YF of 1800 in Cádiz occasioned much controversy during the first decades of the 19th century, relating to the possible causes of its appearance and propagation. The main objective of this paper is to study the climatic conditions prevailing during this YF epidemic, as well as the scientific discussion of its origins, when meteorology was considered as an ancillary science of medicine. As we will see, the medical controversy was an incentive to deepen the meteorological knowledge of the time.

The outline of the article is as follows: Section 2 describes the study area and the main aspects of the YF epidemic of 1800, while Section 3 describes the weather information obtained from diverse documentary sources. Section 4 shows the main arguments of the controversy and their implications for meteorological science. Lastly, Section 5 summarizes some of the findings and research perspectives for the future.

## 2. The YF Epidemic in Cádiz in 1800

Cádiz (36°29′ N, 6°15′ W; Figure 1) is located in the southwestern Iberian Peninsula. The city is set on an island connected to the mainland by a long isthmus that forms a wide, protected inlet. The Cádiz Bay includes wetlands, beaches, pine forests, and scrub areas. Three rivers flow to the sea—the Guadalete, Iro, and Salado rivers—with many channels among them [16]. In addition, during the second half of the 18th century, there was an increase in salt flats near the sea [17]. The climate of Cádiz is characterized by warm temperatures during the summer, relatively mild winters, and high humidity levels.

Because of the technical difficulties that large ships had getting to Seville along the River Guadalquivir, Cádiz became the port that held the shipping monopoly between Spain and its colonies in America from the beginning of the 18th century onwards [18]. There, in 1800, in spite of war conflicts, 18 ships arrived at Cádiz from America, and 19 ships went from Cádiz to America [19]. The city of Cádiz underwent an intense period of economic and social development during the 18th and the beginning of the 19th century. It has been estimated that the population of Cádiz in 1800 was around 70,000 inhabitants [20].

The disease began from 10 to 15 August, after the arrival of a ship from La Havana (Cuba), with three sailors who had died from YF. The disease rapidly spread to the city and other areas of

southwestern Spain, reaching Seville. From August to the beginning of November, in Cádiz, around 7000 people died—around 10% of the population. It has been estimated that only 13.55% of the population was not affected by the illness [19]. The epidemic finished on 12 November [21].

This epidemic was not a unique YF epidemic in Spain during the first decades of the 19th century (Figure 1). YF epidemics have been recorded in Málaga in 1804 and 1813, Cartagena in 1810, Cádiz in 1813 and 1819, and Barcelona in 1821 [22]. In all of the cases, the epidemic had a clear seasonal character, with a peak in the summer months.

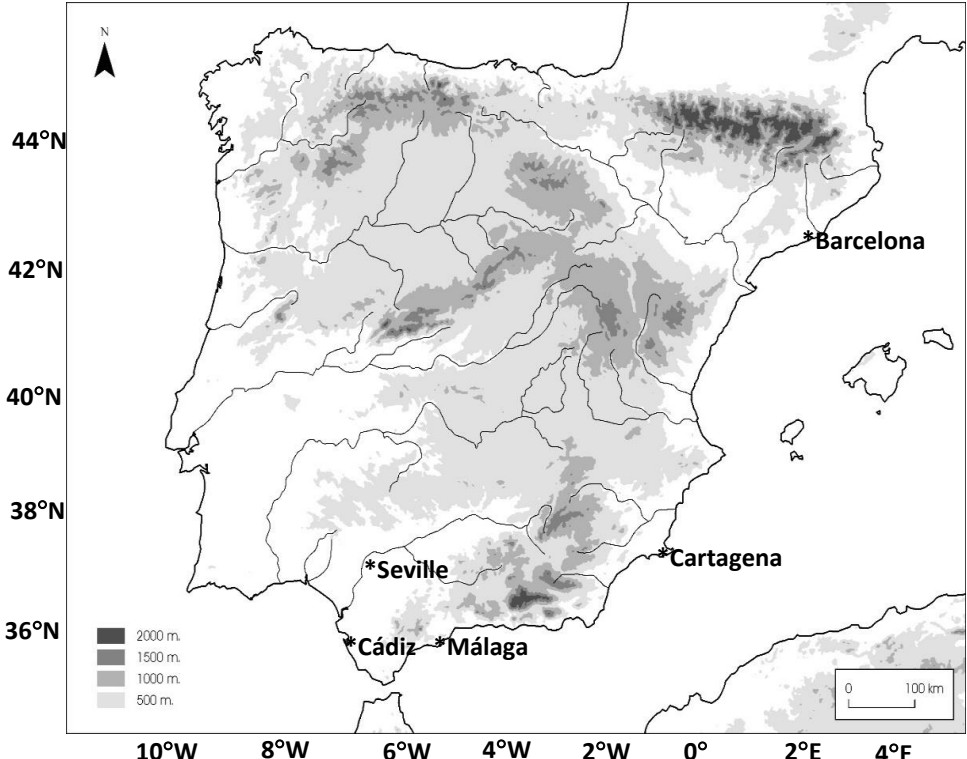

**Figure 1.** A map of Spain indicating the main cities with yellow fever (YF) epidemics at the beginning of the 19th century: Cádiz (1800, 1813, and 1819), Seville (1800), Málaga (1804 and 1813), Cartagena (1810), and Barcelona (1821).

## 3. The Climate of Cádiz in 1800

Various documentary sources [23–25] report that in 1800, a long and humid winter was succeeded by a very hot summer, with constant and sultry east wind in July and August. The *Medical News* [25] noted a temperature of 85° Fahrenheit (29 °C) in mid-July, an anonymous author [23] indicated that a maximum temperature of 88 °F (31 °C) was reached on 19 August, and the physician P.M. González [24] recorded a temperature of 90 °F (32 °C) in August.

The main data source for the climatic conditions in Cádiz during the year 1800 is the text by J.M. Aréjula [26], where he studied the YF epidemic of 1800 after compiling meteorological data on a daily scale from 1799, 1800, and 1803. This author yields daily data of the temperature, pressure, wind direction and state of the atmosphere taken daily at 12 h during 1799 and 1800. The author conveys that the instruments were located at 12 feet above the sea level (around 4 m). The barometer used is unknown. After a preliminary inspection, it was evident that the barometer in use was of French origin, using French inches, lines, and fractions of a line. The thermometer was an English instrument, with data expressed in degrees Fahrenheit and fractions of a degree. There is no suggestion that they were taken in outdoor shaded locations or indoors. However, the analysis of the day-by-day variability of temperature shows that the thermometer was probably indoors, or strongly conditioned by the building structure. Wind direction was recorded on a 16-point compass. Finally, the state of the

atmosphere is a qualitative description of rain, cloudiness, and other events. Data corresponding to the year 1803 cover the period from 1 January to 17 October. In this case, observations were taken three times daily: in the morning, at noon–afternoon, and in the evening. The author conveys that the series is interrupted on the 17 October because he had to travel to other city in southern Spain (Málaga) for professional duties. The instruments used by Aréjula and their exposure conditions are unknown. In this case, air pressure data seem to be consistent with English system and are expressed in inches, lines, and tenths of a line. Temperature data are expressed in degrees Reamur and fractions of a degree. Again, we do not know if the instruments were exposed to the open air or, conversely, kept indoors, but the thermometer was placed indoors in the house of the author, located in the Old City. Wind direction was recorded on a 16-point compass. Aréjula includes information about the wind strength, with terms used in the ship logbooks of the time. The description of the state of the atmosphere consists of a list of very different qualitative terms, including references to rainy days, cloudiness, wind, storms, etc. This data source has been studied previously [27,28], and the data have been digitized and they are available in the "Early Meteorological Observations in Southern Spain" database (EMOSSv2, [29]). Unfortunately, a more detailed description of metadata (the types of instrument and producers) is absent, and this problem must be taken into account when analyzing these data and their uncertainties.

Figure 2 shows the monthly means of daily maximum temperatures (TX) from January 1799 to December 1800 in Cádiz according to the data given by Aréjula. These data are compared with those corresponding to the modern reference period 1961–1990 [30]. The period 1961–1990 is interesting because it includes different climatic conditions, with a shift towards warmer conditions around 1980, globally and regionally [31]. However, other reference periods were proved (i.e., 1971–2000) and the results of the comparison were similar. Error margins consider the uncertainty due to the instruments and measurement units, and the standard error in the estimation of the mean value [32]. Historical data reproduce the annual cycle of temperature, but with values lower than those of the reference period, particularly in 1799. This difference has been assigned to the general coldest conditions being during the Dalton Minimum of solar activity [33]. This period belongs to the so-called Little Ice Age (LIA). The coldest period during the LIA is considered to be the 17th century, when a social and economic crisis spread globally [34]. The end of the LIA is normally dated to the mid-19th century, at the beginning of the industrial period [35], although according to dendroclimatological studies [36–38], the end of the LIA in the Iberian Peninsula can be dated to the 18th century, in the 1770s. However, cold and wet conditions in southern Spain continued until at least the mid-19th century. The main impacts of these conditions were the frequent rainfalls and floods of the Guadalquivir River in Seville during the 1780s [28], and the so-called "year without summer" in 1816, with temperature anomalies within the −1.2 to −2 °C range [39]. However, the values corresponding to 1800 are very similar to modern values, ranging between 25.4 and 28.4 °C in July 1800 (27.4 °C in 1961–1990), and between 25.1 and 28.1 °C in August 1800 (27.9 °C in 1961–1990). This result underlines the marked variability of the climate in the Iberian Peninsula during the LIA [14,37]. Under the general cold conditions of this period, it is the logical qualitative perception of contemporaneous authors that it was a very hot summer. These conditions were also noted in other documentary sources, such as the weekly reports of the newspaper *Correo Mercantil de España y sus Indias* (CMEI [40]), in which information on "excessively hot weather" during July and August 1800 in the southern Spanish provinces may be found [41]. The periodical appeared twice a week, on Mondays and Thursdays, from 1792 to 1808, with alternating information from the northern and southern provinces each day, thus providing a weekly summary of the general conditions (economic, agricultural, and meteorological) across the country. All the editions began with a report on agriculture, qualitatively describing general weather conditions and indicating the grain prices.

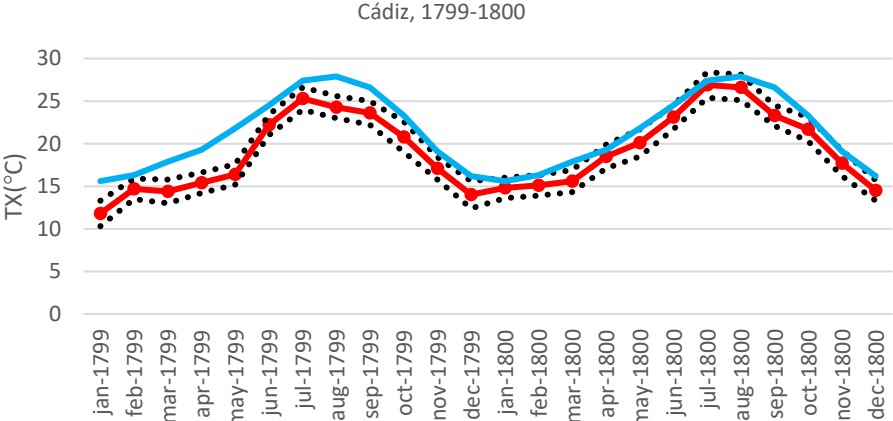

**Figure 2.** Red: monthly mean values of daily maximum temperatures from January 1799 to December 1800. Dotted black line: error margins. Blue: monthly mean values of daily maximum temperatures for the reference period 1961–1990.

Figure 3 shows the monthly number of rainy days (RDs) according to the Aréjula data. Again, these data are compared with the mean value corresponding to the reference period in 1961–1990. This variable is important, because it indicates the degree to which a constant supply of moisture is available in order to provide proper breeding grounds [13]. Here, it can be seen that from December 1799 to March 1800, the RDs were higher than the reference period values, confirming the information about wet conditions during the months prior to the summer of 1800. The information about continuous and strong rainfalls until the end of March is also included in the reports of CMEI. There is a one-month lag between the peak rainfall and the peak river flow, as the rivers are recharged by surface runoff and groundwater flow from the drainage basin [6,42]. Therefore, the moisture content of wetlands in the Cádiz Bay probably increased during the spring, increasing the availability of mosquito breeding habitats, and thus the size of the mosquito population. In fact, there are references in the literature of the time [22,26,43] to standing waters and their role as a source of miasmas, responsible of the appearance of illness (miasmas was understood as a kind of corrupt or pestilent air that emanated from putrefactive bodies and spread infectious diseases).

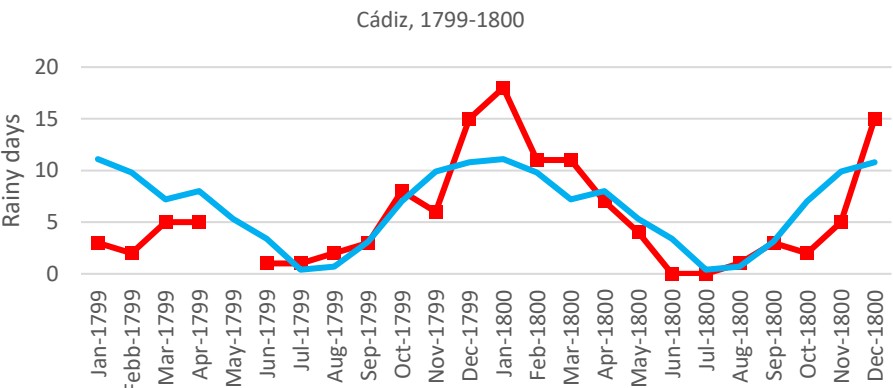

**Figure 3.** Red: monthly rainy days in Cádiz from January 1799 to December 1800. Blue: the monthly mean value of rainy days in Cádiz for the reference period 1961–1990.

Figure 4a shows the prevailing wind directions during August 1800, with a clear predominance of east winds (29%). These data are compared with the mean value of the wind frequencies in August in Cádiz corresponding to the period 1788–1795 (Figure 4b [44]), with 16.4% being east winds. Although the wind rose of 1800 is more detailed than that of 1788–1795 (16 versus 8 compass points), the differences are clear, showing the anomalous character of August 1800, namely: west winds were absent in 1800 (47% in the previous decade), and east winds during August 1800 (29%) were almost

double the mean value corresponding to the previous decade (16.4%). This fact was perceived by contemporaneous authors, who noted that during August and September, "the east wind persisted more than forty days" [24]. The predominance of east winds in the Seville province is also included in the weekly reports of CMEI. The east wind in Cádiz in summer is hot and dry, and it contributes to the reinforcement of the warm thermal sensation. These conditions are provoked by the predominance of anticyclonic conditions over the Iberian Peninsula, as reflected in Figure 4c, which shows the independent reconstruction of the mean value of the sea level pressure field during August 1800 [45]. This reconstruction was made from a multi-proxy approach of past climate in Europe [46].

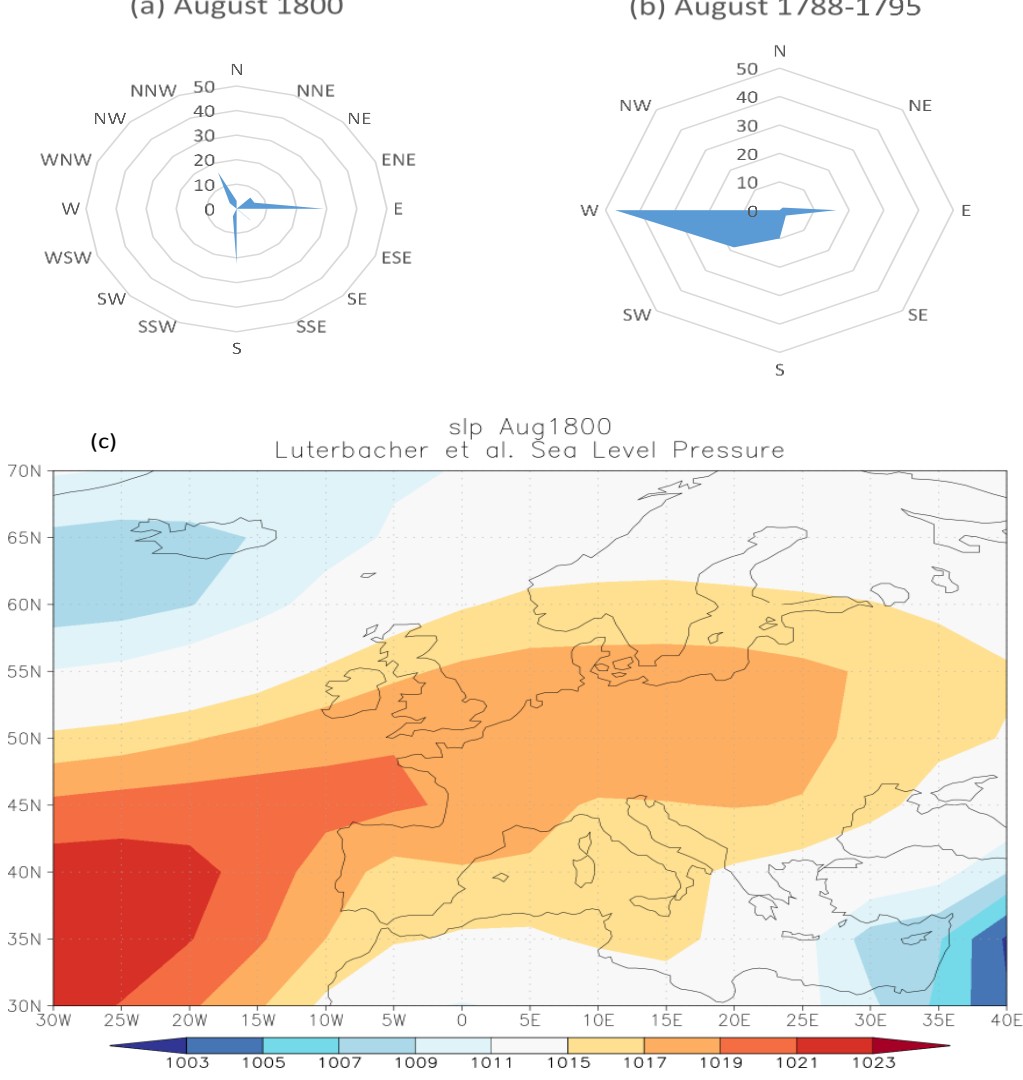

**Figure 4.** (**a**) A wind rose corresponding to August 1800. (**b**) The August wind rose for the period 1788–1795; (**c**) The sea level pressure (SLP) mean field according to Luterbacher et al.'s reconstruction [45].

Rainy conditions between December 1799 and March 1800 and warm weather during the summer months may have contributed to the YF outbreak, by promoting the reproduction, survival, and propagation of the *Aedes aegypti* mosquito. Similar conditions were recorded in other YF epidemics. Therefore, in Málaga in 1804, the winter was mild and rainy, with maximum temperatures of around 17 °C; the spring was mild and very wet; and the summer was very warm, with maximum temperatures of 36 °C "in the shadow" [47]. The YF epidemics in Cádiz in 1819 also occurred during a very hot summer, with a high frequency of east winds [44]. In a previous study on climate conditions in southern Spain during the Dalton Minimum [41], it was found that wet springs and warm and dry summers prevailed

during this time. Therefore, it seems that conditions appropriate for the development of mosquitoes and the appearance of diseases prevailed. This situation could have led some authors to consider the YF as endemic in southern Spain, which "resembled the ordinary climate of the Antilles" [26].

## 4. The Controversy

At the beginning of the 19th century, physicians did not know the role of mosquitoes in the transmission of diseases. They followed the neo-Hippocratic hypothesis. According to this medical paradigm, illness, epidemics, and public health were related to environmental conditions, in particular, to the variability of meteorological variables. The controversy focused on the origin and character (contagious or not) of the disease.

The first document on the YF in Cádiz in 1800 is an anonymous book published at the end of that year [23]. According to the author, we must "focus on heat and moisture as main factors causing the corruption". Afterwards, he indicated that the high values of temperature measured in Cádiz during August 1800 were "almost similar to the highest heat measured in the globe", in reference to the Caribbean climate. In a work published in 1821, the English physician R. Jackson [48] assigned the origin of epidemics to "causes inherent in the physical qualities of the soil and atmosphere", and T. O'Halloran, in his study published in 1823 [49], underlined the strong seasonality of the illness ("it never appears before the end of July or the beginning of August, and it begins to abate in violence in October, and has never been observed to continue till November or December") and that "its presence has been generally preceded by an unsual state of the atmosphere". The conclusion is that the prevalence of a peculiar state of the atmosphere is the "essential cause on which the formation and propagation of febrile epidemics depend".

However, the fact that epidemics prevailed in maritime towns, with traffic between America and Spain, led to considering their origin to be foreign importation. The physician P.M. González [24] indicated the arrival of ships from America just before the beginning of the illness, and the similarities of the illness with diseases in the Antilles. He compares Cádiz with other cities in southern Spain, with warmer and drier climates, to refute the previous thesis, considering the importance of local factors in climate and public health. In this text, the heat is considered as a "concause, able to active the contagion, increasing its propagation and virulence". The YF of Cádiz in 1800 contributed to the development of a genre of scientific literature—medical topographies [50]. These reports studied local conditions (geographical, hydrological, and climatological) to analyze the public health in a concrete city (see, for example, [51]).

Aréjula [26] denied the role of high temperatures as a main causal factor. His argument was based on a comparison of the maximum temperatures of other years. Figure 5 shows the maximum temperatures recorded in Cádiz from 1789 to 1803. The maximum temperature recorded in 1800 was 30.6 °C on 19 August. The argument by Aréjula was that in other years with temperatures similar or higher than those in 1800 (32.2 °C on 27 July 1790 and 31.7 °C on 7 July 1803), there was not an epidemic. A similar argument was used by the physician J. Mendoza in his description of the YF in Málaga in 1804 [47]. The maximum temperature in this city in 1804 was 36 °C on 13 August; meanwhile, on 9 July 1808, the thermometer reached 45.3 °C (thermometer in the shadow), and there were no health problems that year.

From a meteorological point of view, the main interest of this discussion is that these measurements were taken at different observatories and under different exposure conditions. Measurements from 1789 to 1794 were taken in the Royal Observatory of Cádiz, located 26.7 m above sea level. Measurements from 1799 and 1800 were taken in the Observatory of La Isla, located in another site of the city, at 40.7 m above sea level. The medical controversy consisted of a methodological discussion on the influence of exposure conditions on instrumental records. According to the anonymous author, the thermometer located in the Royal Observatory of Cádiz was unprotected from the sunrays; meanwhile, the thermometer in La Isla was in the shadow. As a consequence, the thermometers from 1789 to 1794 were influenced by sunrays, and their measurements were biased, recording values higher

than the correct values. Therefore, 1800 would be the warmest year in the complete series. Aréjula refuted this argument, adding that the temperature recorded in La Isla in 1803 was directly comparable with the temperature recorded in 1800. The efforts of the anonymous author to describe the differences between the measurements taken outdoors and indoors, and in different seasons of the year, were not countered by Aréjula, because this problem was outside of the scope of his book. However, these efforts indicate that authors were aware of these methodological and practical problems.

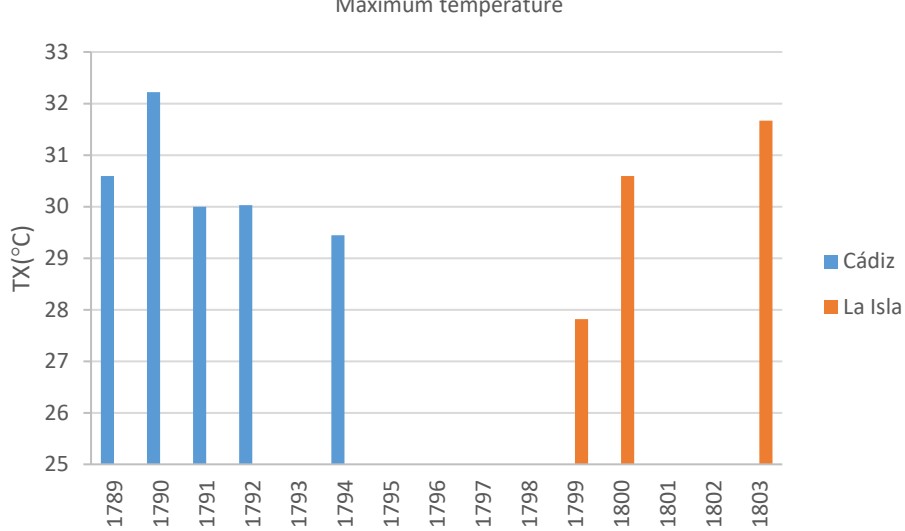

**Figure 5.** The maximum temperatures recorded in Cádiz (blue) and La Isla (orange) from 1789 to 1803.

Aréjula was a neo-Hippocratic physician. As a consequence, he had to develop a methodology including environmental factors in the explanation of epidemics. Thus, he spoke on external causes (contagion), underlying causes (vulnerability of the individual), and concauses (meteorological factors). Today, we would speak about vulnerability, that is, the tendency or predisposition to be adversely affected by climate-related health effects, which encompasses three elements, namely exposure, sensitivity or susceptibility to harm, and the capacity to adapt or to cope [2]. Note the similarities between historical and modern concepts.

According to pro-contagion authors, the role of high temperatures was to promote the appearance of contagious miasmas. Water masses were the main source of these miasmas. Thus, they recommended cleaning mains and streams, and desiccating areas with water-stilling waters [22]. According to anti-contagion authors, on the contrary, "marshy situations are undoubtedly insalubrions, but they are not the sole causes of epidemic disease" [49]. Pro-contagion authors underlined the need for establishing quarantines, and anti-contagion authors indicated "the irregularitity of the malady, jumping from point to point and leaving intermediate places untouched" ([49], note that this last argument suggests the random flight of mosquitoes).

Anti-contagion authors argued a cause-effect relationship between diseases and climate, attributing the disease to warm conditions in the summer and the specific geographical characteristics of the region. Their criticism of the person-to-person contagion model was correct. On the other hand, pro-contagion authors included environmental factors in a more complex explanation, where the combined action of environmental conditions, contagion, and the susceptibility of individuals contributed to the appearance and propagation of the disease. Pro-contagion authors were correct in assigning the origin of the disease to foreign importation. Neither pro- nor anti-contagion authors knew the role of mosquitoes. However, their discussions contributed to the deepening of meteorological science, including the analysis of local climates and methodological problems associated with instrumental measurements [52].

## 5. Conclusions

The main conclusions of this work are the following:

- Climate conditions in Cádiz during 1800 (wet winter and spring, warm summer) were appropriate for the development of mosquitoes and the outbreak of the YF epidemic.
- Differences in measurement sites and the exposure conditions of the meteorological instruments precluded the comparison of different measurements and promoted the scientific discussion of the meteorological conditions during the summer of 1800.
- The medical controversy surrounding the origin and propagation of the epidemic contributed to the deepening of meteorological science in Spain, promoting local climate studies and the analysis of problems related to the exposure conditions of meteorological instruments.

The study of past diseases contributes to the knowledge of complex relationships between climate and human health. The example studied here may help to understand other past epidemic events in Spain, for instance, the pest epidemics during the 17th century. On the other hand, it allows for obtaining an overview of the relationship between medicine and meteorology during the studied period. Although meteorology was an auxiliary science of medicine, it was promoted by the neo-Hippocratic paradigm, which encouraged observational studies and instrumental measurements.

**Funding:** This research received no external funding.

**Acknowledgments:** The author wishes to express his gratitude to the anonymous referees for their useful comments.

**Conflicts of Interest:** The author declares no conflict of interest.

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
