# Peer review of "The Influence of Meteorological Conditions on the Yellow Fever Epidemic in Cádiz (Southern Spain) in 1800: A Historical Scientific Controversy"

_atmosphere, doi:10.3390/atmos11040405_

Round 1

Reviewer 1 Report

The objective of the reviewed paper was to recognize the climatic conditions bevor and during a yellow fever epidemic in Cádiz (southern Spain) during the last months of 1800, as well as presenting the historical scientific discussion on its origins. In that discussion, the meteorology was considered as the ancillary science of medicine.

The reviewed article falls within the thematic scope of historical meteorology, climatology, and biometeorology as well as in the field of medical climatology. The purpose of the study was set correctly. The authors based their considerations on well-chosen historical literature. The pre and post epidemic meteorological conditions were presented correctly. Obtained in the paper results are interesting and complement the knowledge about the climate at the turn of the eighteenth and nineteenth centuries in Cádiz, but also present the views prevailing at that time about the causes of the yellow fever epidemic.

Comments:

  1. In the description of Figure 2, there should be dotted black lines.

Reviewer 2 Report

This is a nice speculative paper of  historiographic and historical interest. It doesn't set out to show its thesis statistically, that the Yellow fever epidemic in Cadiz in 1800 was connected to weather factors, but it would nevertheless strengthen the paper to look a bit up from its focus period and look at the decades before and after and try to quantify how special was time period was weather-wise. In general, though, a nice readable paper.

Reviewer 3 Report

This is an interesting research paper dealing with the influence of meteorological conditions on the yellow fever epidemic in Cádiz, Spain.

However there are some major points to be reflected:

  1. There is a mixture of historical facts and the interpretation from a historical point of view and a current point of view. These three aspects of the research have to be split up very clearly.
  2. Therefore it might be helpful to explain how YF (Aedes aegypti mosquito) biology works today. And then to explain the outbreak and the stop of the epidemic from a historical and a current point of view.
  3. In the must become clearer why the reference period 1961-1990 was chosen.
  4. Chapter 4, line 317However, their discussions contributed to deepening  meteorological science, including the analysis of local climates and methodological problems associated with instrumental measurements.

    Such a statement must be referenced by the corresponding literature in order to proof such findings.

  5. Chapter 4 indicates well the problem of changing measuring sites, but there is a need also of knowing something about the instruments (what kind of type, who produced them, was is all the time the same type)

  6. Chapter 5 is not in line with the findings of chapter 4. For example it is crucial to conclude that meteorological measurements were not comparable due a change of measurements sites.

Round 2

Reviewer 3 Report

The author provided a new version of the publication that responds to the points raised by the reviewer. In fact the publication is now better readable and understandable and the quality has been accordingly improved.